# Patient-Derived Xenograft Models in Cervical Cancer: A Systematic Review

**DOI:** 10.3390/ijms22179369

**Published:** 2021-08-29

**Authors:** Tomohito Tanaka, Ruri Nishie, Shoko Ueda, Shunsuke Miyamoto, Sousuke Hashida, Hiromi Konishi, Shinichi Terada, Yuhei Kogata, Hiroshi Sasaki, Satoshi Tsunetoh, Kohei Taniguchi, Kazumasa Komura, Masahide Ohmichi

**Affiliations:** 1Department of Obstetrics and Gynecology, Educational Foundation of Osaka Medical and Pharmaceutical University, 2-7 Daigakumachi, Takatsuki, Osaka 569-8686, Japan; ruri.nishie@ompu.ac.jp (R.N.); shouko.ueda@ompu.ac.jp (S.U.); shunsuke.miyamoto@ompu.ac.jp (S.M.); gyn152@osaka-med.ac.jp (S.H.); gyn130@osaka-med.ac.jp (H.K.); shinichi.terada@ompu.ac.jp (S.T.); yuhei.kogata@ompu.ac.jp (Y.K.); hiroshi.sasaki@ompu.ac.jp (H.S.); satoshi.tsunetoh@ompu.ac.jp (S.T.); m-ohmichi@ompu.ac.jp (M.O.); 2Translational Research Program, Educational Foundation of Osaka Medical and Pharmaceutical University, 2-7 Daigakumachi, Takatsuki, Osaka 569-8686, Japan; kohei.taniguchi@ompu.ac.jp (K.T.); kazumasa.komura@ompu.ac.jp (K.K.)

**Keywords:** gynecological malignancy, cervical cancer, patient-derived xenograft, PDX, mouse model

## Abstract

Background: Patient-derived xenograft (PDX) models have been a focus of attention because they closely resemble the tumor features of patients and retain the molecular and histological features of diseases. They are promising tools for translational research. In the current systematic review, we identify publications on PDX models of cervical cancer (CC-PDX) with descriptions of main methodological characteristics and outcomes to identify the most suitable method for CC-PDX. Methods: We searched on PubMed to identify articles reporting CC-PDX. Briefly, the main inclusion criterion for papers was description of PDX created with fragments obtained from human cervical cancer specimens, and the exclusion criterion was the creation of xenograft with established cell lines. Results: After the search process, 10 studies were found and included in the systematic review. Among 98 donor patients, 61 CC-PDX were established, and the overall success rate was 62.2%. The success rate in each article ranged from 0% to 75% and was higher when using severe immunodeficient mice such as severe combined immunodeficient (SCID), nonobese diabetic (NOD) SCID, and NOD SCID gamma (NSG) mice than nude mice. Subrenal capsule implantation led to a higher engraftment rate than orthotopic and subcutaneous implantation. Fragments with a size of 1–3 mm^3^ were suitable for CC-PDX. No relationship was found between the engraftment rate and characteristics of the tumor and donor patient, including histology, staging, and metastasis. The latency period varied from 10 days to 12 months. Most studies showed a strong similarity in pathological and immunohistochemical features between the original tumor and the PDX model. Conclusion: Severe immunodeficient mice and subrenal capsule implantation led to a higher engraftment rate; however, orthotopic and subcutaneous implantation were alternatives. When using nude mice, subrenal implantation may be better. Fragments with a size of 1–3 mm^3^ were suitable for CC-PDX. Few reports have been published about CC-PDX; the results were not confirmed because of the small sample size.

## 1. Introduction

Cervical cancer, caused by persistent infection of human papillomavirus (HPV), ranks the second common cause of cancer-related deaths worldwide [1]. For early-stage patients, surgical resection or radiotherapy provides a satisfactory prognosis. However, the patients with advanced-stage or recurrent disease have dismal prognosis [2]. An effective therapy and biomarker determinates of therapeutic response for refractory cervical cancer is expected to be found. In colorectal cancer, dual HER2 blockade with trastuzumab and lapatinib led to inhibition of tumor growth in patient-derived xenograft (PDX) of HER2-amplified tumor [3]. Phase 2 clinical trial proved the effectiveness of this therapy in treatment-refractory patients with HER2-positive metastatic colorectal cancer [4].

Many researchers have used cell lines in cancer research, and various cell lines have been established over the last several decades. The benefit of cell lines is that the cells have an identical arrangement of genes and similar reactions through molecular pathways in some situations, and they are easy to use in cancer research. Accordingly, current drug development has been mainly pursued using cell lines. However, many studies have reported that drug susceptibility of cell lines is not sufficiently reflected in human patients [5,6,7]. With the development of molecular and genomic analyses, researchers have focused on more detailed information in individual cancer cell genes and the tumor microenvironment. As a result, new animal models called PDXs have been established in which tissue resected during surgery or biopsy is grafted into immunodeficient mice. PDX models have similar pathological features to the primary tumor and are used as animal experiments for evaluation of drugs, biomarker identification, and precision medicine. The sensitivity of PDX models to drugs is more similar to clinical drug responsiveness because these models have similar genomic features and microenvironment status to the primary tumors [8,9,10,11,12]. The most effective drugs can be identified by screening some anticancer agents in PDX models. However, this method is not easy to apply because the engrafted tissue does not grow in all cases, and several months may be needed to obtain drug sensitivity data from PDX models [8,11,12,13]. A PDX cohort with gene information and drug sensitivity data has been suggested. It could be a useful database for anticancer agent development and therapy for cancer patients. A large PDX cohort could cover almost every stage and grade of histological subtype. Furthermore, clinical information, such as prognosis and drug sensitivity, can also be recorded [14,15,16,17]. Briefly, all materials and information from cancer patients and PDX models are stored in biobanks and data banks. The materials include all samples obtained from patients or PDX models, such as blood, urine, discharge, and tumors. The information also includes data, such as clinicopathological findings, genomic analysis findings, and drug sensitivity data. These materials and information in biobanks or databanks are intended for use in precision medicine and development of anticancer agents; this platform allows many researchers to share all types of information and conduct experiments with PDX that reflect the characteristics of the primary tumor (Figure 1) [14,15,16,17].

PDX models of several cancers have been reported, including cancers of the colon [18], stomach [19], breast [20], pancreas [21,22], lung [23,24], liver [25], kidney [26], bladder [27], uterus [14,28,29,30,31,32,33,34,35,36,37,38,39,40,41,42], and ovary [43,44,45,46,47]; however, several problems remain. For example, what should be used as the starting material, tissue fragments or tumor cells? Where should the materials be implanted? What is the success rate of each method? Is there any advantage or disadvantage to each method? Are there any alterations following retransplantation? This review compiles the information on current methods in cervical cancer PDX (CC-PDX) models.

## 2. Results

### 2.1. Study Selection

The literature search according to the guidelines of the preferred reporting items for systematic reviews and meta-analyses (PRISMA) statement [48] yielded 774 articles. We also identified 296 articles in the references of the selected articles. Among these, 182 duplicate articles were removed. The inclusion and exclusion criteria were applied to the remaining 888 articles, and the full text of 45 articles was read. Finally, 10 articles were selected for the systematic review. Figure 2 illustrates the process of search and selection. The studies were published between 2004 and 2021.

### 2.2. General Information about CC-PDX Models

Table 1 and Table 2 shows information about the selected studies for the current review. The studies were performed in seven countries: Germany (one article), Canada (one article), Japan (two articles), Korea (three articles), Norway (one article), Australia (one article), and China (one article). Five different mouse models were used: nude, severe combined immunodeficient (SCID), nonobese diabetic (NOD)/SCID, NOD SCID gamma (NSG), and NOD-Prkdcem26ll2rgem26Nju (NCG). The histological type of the primary tumor was squamous cell carcinoma or adenocarcinoma in most cases. The procedure for tumor acquisition was provided in eight studies, and was mostly described as radical hysterectomy or biopsy with no details. The time between tumor resection and engraftment varied from immediately to 12 h and was reported in five studies. The implanted tissue fragment varied in size from 1 mm to 5 mm, and one article described the fragment used as minced. The most common transplantation site was the subcutaneous (s.c.) tissue in the dorsal region, followed by the subrenal capsule. Two orthotopic implantations were identified in the included studies. In most cases, the graft was directly implanted using a skin cut. In two articles, minced tumor fragments were injected into s.c. tissue. The duration until visual confirmation of tumor growth was mentioned in seven studies, ranging from 10 days to 1 year. The number of donor patients ranged from 1 to 33. Among 98 donor patients, 61 CC-PDX were established, and the overall success rate was 62.2%. The success rate was reported in six articles and varied from 0% to 75%. The validation methods and parameters used to characterize the tumors of PDX and donor patients are shown in Table 3. All studies reported histological comparisons with the original tumors. Driver gene mutations, gene expression, copy number variations, and proteomics were not evaluated in most studies. Gene expression and copy number variations of HER-2 were evaluated in only one study. Immunohistochemistry was performed in five studies, mainly using anti-p16 antibodies.

### 2.3. Quality Assessment

A previously created model validation tool [49] has been extended for quality assessment of the studies included in this systematic review (Figure 3). All papers described the model in detail, with information about the tissue tracked/origin, and confirmation that the xenograft was from a specific patient, with information about the tracked/proven tissue of origin and confirmation that the xenograft was derived from a donor patient. Most articles included an ethical statement and a histological comparison of the xenograft with the patient tumor. The authors did not adequately describe the routine maintenance of the model, the preparation of further models for experimentation, and the patient with respect to response to standard care or treatment.

### 2.4. Treatment of Specimens

In most studies, the tumor tissues obtained from cancer patients were washed with saline, placed in cell culture medium, stored on ice, and transplanted into the animals immediately [28,29,34,35,36]. Usually, the tumor was cut into fragments and implanted into immunodeficient mice. The growth of the implanted tumors was regularly monitored. When the size of the tumor reached about 1000 mm^3^, the tumor tissue was harvested and stored for analysis and the next stage. A duration of 3 to 6 months was required for sufficient tumor growth in most cases [28,29,30,31,32,33,34,35,36].

### 2.5. Mouse Strain

In CC-PDX, nude mice were the most used models. However, the engraftment rate was higher when severe immunodeficient mice rather than nude mice were used [28,29,30,31,32,33,34,35,36,37]. When nude mice were used, transplantation into the subrenal capsule may result in a high success rate.

### 2.6. Site of Transplantation

Several transplantation sites have been reported for CC-PDX, including s.c. tissue [28,30,33,34,36], subrenal capsule [31,32,35,37], and orthotopic tissue [29,30]. S.c. transplantation was the most common procedure because of its simplicity and ease of confirming tumor implantation; however, metastasis to other organs is rare. The collapsed tumor may be injected into s.c. tissue [28,36]. After incising the skin, the tumor fragment may be placed directly on s.c. tissue [28,30,33,34]. The renal capsule can be used for low-grade tumors as well as for normal tissue. Although the procedure is not easy, the blood supply for tumor growth is increased in the renal capsule, and a high transplantation rate is expected. Mice were placed on their side, and a 2-cm incision was made on the opposite side of the skin at the waist. Access to the peritoneal cavity was created through an incision in the abdominal wall above the kidney. After exteriorizing the kidney, the renal capsule and the space beneath the kidney capsule were opened. The tumor fragment was then inserted [31,32,35,37]. Orthotopic xenotransplantation is commonly performed because it more accurately recreates the environment of the tumor. After making an incision in the skin and exposing the abdominal cavity, the tumor fragment was attached to the cervix [29,30].

### 2.7. Engraftment Rate of Each Study

Among 98 donor patients, 61 CC-PDX were established, and the overall success rate was 72.2%. Hiroshima et al. transplanted HER-2-positive cervical cancer resected from one patient into the s.c. region or cervix of several nude mice, and the transplantation rate was 70–75% [30]. Chaudary et al. implanted 1- to 2-mm tumor fragments removed from 33 cervical cancer patients into the cervix of SCID mice and found a 48% transplantation rate. After orthotopic transplantation, an average of 3 to 4 months were required for the first palpable xenograft to develop [29]. Hoffmann et al. transplanted 3- to 5-mm tumor fragments excised from six cervical cancer patients into the s.c. region of SCID mice but did not observe any tumor viability. Then, minced tumors excised from seven cervical cancer patients were injected into the s.c. region of SCID mice with a transplantation rate of 70%. Six to 8 weeks were required for implanted tumor tumors to be visible or palpable. No difference in the transplantation rate was observed between squamous cell carcinoma and adenocarcinoma [28]. Larmour et al. transplanted 1-mm^3^ tumor fragments excised from 14 cervical cancer patients into the renal capsule of NSG mice and found a 71.4% viability rate, and they also reported that the mouse stroma did not contribute to reimplantation. Transplantation of 10^6^ cells into the kidneys did not produce tumors, regardless of the presence of mouse cells. The harvested xenograft size limited the possibility of reimplantation. The authors also showed the survival and growth of cervical dysplasia and normal tissue xenografted under the renal capsule [35]. Oh et al. reported transplantation of a 1-mm^3^ fragment into the subrenal capsule of nude mice with a success rate of 66.7% [32]. In the studies with a small sample size, no differences in the success rate were observed in stage, histology, and lymph node metastasis [28,29,32,35]. In summary, 1- to 3-mm^3^ fragments were suitable for CC-PDX. Furthermore, subrenal capsule implantation and using severe immunodeficient mice improved the success rate.

### 2.8. Comparison of the Original Tumor and PDX

Most published studies on CC-PDX show a strong similarity in pathological features between the original tumor and the PDX model. Overexpression of p16, resulting from functional inactivation of Rb by the HPV E7 protein, is frequently observed in cervical cancer. In addition, the immunohistochemical features of these proteins are inherited by PDX from primary tumors. Hiroshima et al. constructed a PDX model of HER-2-positive cervical cancer by implanting tumor fragments into the s.c. tissue and cervix of nude mice. No metastasis was observed in the s.c. PDX mice. On the other hand, in the cervical orthotopic PDX model, peritoneal dissemination and metastasis to the liver, lung, and lymph nodes were observed. The authors showed that orthotopic and s.c. xenograft tumors and metastases were stained with anti-HER-2 antibodies, reflecting the histological features of the original tumor [30]. Chaudary et al. compared the epithelial and stromal components of the tissue from the original biopsy and the xenograft model. The proportion of stroma tended to increase in the early stage and decrease in the later stage, and was low (less than 10%) after the fifth stage. When assessed using the average time between passages, the decrease in stromal components in later passages was paralleled by an increase in the rate of tumor growth. The epithelial and stromal components were also evaluated by immunohistochemistry for smooth muscle actin (SMA), collagen IV, cytokeratin, CD31, lymphatic vessel endothelial hyaluronan receptor 1 (LYVE1), EF5, carbonic anhydrase-9 (CA-9), and Ki67. CA-9 and EF5 for hypoxia markers were highly expressed in epithelial components with increasing numbers of passages. Similarly, CD31 for vascular staining was highly expressed in stromal components with increasing numbers of passages. Expression of Ki67 in epithelial components and LYVE1 in both components was significantly increased with the number of passages. CA-9 and EF5 were more greatly altered in epithelial components than in stromal components. The expression of CD31 and LYVE1 was much lower in epithelial components than in stromal components. The Ki67 index was much lower in stromal components than in epithelial components. For all markers except collagen IV, a strong correlation was observed between the primary biopsy specimens and xenografts in passage three [29]. Larmour et al. showed similar morphological findings by H&E staining and immunohistochemical features for p16 and HPV between primary tumors and serially passaged xenograft tumors. Hoffmann et al. showed that the expression of epidermal growth factor receptor and p16 was preserved through passage of PDX [35]. Oh et al. established a HER-2-amplified CC-PDX. They confirmed HER-2 overexpression in the original tumor and in the serially passaged PDX. Co-administration of trastuzumab and lapatinib in HER-2-overexpressing PDXs significantly inhibited tumor growth compared with the control [32].

## 3. Discussion

### 3.1. Treatment of Specimens

In general, tumor tissues obtained from cancer patients should be transplanted into the animals immediately. A prolonged implantation time causes tissue metabolism, resulting in a lower engraftment rate [28,29,34,35,36,50]. The volume of sample should be a certain amount for engraftment [11]. Hoffmann et al. reported that no engraftment was detectable when using tumor pieces that were solid 3- to 5-mm cubes [28]. Tissue fragments obtained by biopsy showed a higher engraftment rate than isolated cells or circulating tumor cells in lung cancer [51].

### 3.2. Mouse Strains

Transplantation rates are important in situations where limited funds or limited specimens are used. Successful transplantation depends on several factors, including the type of recipient mouse, the implantation site, the implantation method, and the size of the fragment. In CC-PDX, nude mice are the most used models. However, the engraftment rate is higher when severe immunodeficient mice rather than nude mice are used [28,29,30,31,32,33,34,35,36,37]. Nude mice without a thymus due to a mutation were identified for their atrichia appearance in 1962; these mice do not have T cells [52]. It is easy to check for s.c. transplants because these mice are hairless. In 1983, SCID mice were discovered to have lost expression of Prkdc (protein kinase, DNA activated, and catalytic polypeptide) and to lack mature T and B cells [53]. SCID-hu can be used to transplant human fetal liver, thymus, and renal envelope tissue containing T cells [54]. Human T cells can be increased in hu-PBL-SCID mice transplanted with human monocytes in the peritoneum. These mice transplanted with human T cells have contributed to the study of human immunodeficiency virus. However, the transplantation rate and duration of human hematopoietic stem cells remaining in the mice have not been satisfactory because SCID mice have natural killer (NK) cell activity [55]. SCID-Beige is a hybrid of SCID and beige mice with low NK cell activity. However, the transplantation rate and duration of human hematopoietic stem cells are not satisfactory [56,57]. NOD mice suffer from insulin-dependent diabetes mellitus in which the beta cells in the pancreas are destroyed by T cells. NOD mice are also characterized by low activity of macrophages and dendritic cells [58]. NOD/SCID mice, which are a cross between NOD and SCID mice, do not show symptoms of diabetes due to a lack of T cells. These mice suffer from extreme immunodeficiency [59]. NOG [60] and NSG [61] mice are hybrids of NOD/SCID and common γ-deficient mice that do not exhibit NK cell activity. NOG and NSG mice can be satisfactorily transplanted with human hematopoietic stem cells, monocytes, and malignant cells. More recently, humanized mice have been used to develop several immune checkpoint inhibitors. CD34-positive human hematopoietic stem and progenitor cells were injected into NSG mice subjected to whole body irradiation, resulting in reconstituted immune cells. The humanized PDX model can be constructed with an allogeneic immune system partially matched with human leukocyte antigens [62,63,64,65].

NOD/SCID mice are usually used for PDX because of their lack of NK cells. However, the different types of mice described above are used in diverse cancers. This means that use of a suitable type of mouse improves the engraftment rate [11]. For example, nude mice were used for PDX of gastric cancer. Gastric and colorectal cancer patients have higher IgG levels than patients with other cancers. Moreover, patients with advanced gastric cancer have higher IgG levels than those with earlier stage disease. The number of B lymphocytes is normal, but their function is lost in nude mice. These characteristics may be associated with the use of nude mice in gastric cancer PDX models [66]. Nude mice are commonly used for PDX of colon [11], renal [67], and pancreatic cancer [21,68]. Transplantation into the subrenal capsule of SCID mice is usually chosen for prostate cancer PDX because a high success rate of 95% has been reported [69].

### 3.3. Site of Transplantation

S.c. transplantation is the most common procedure because of its simplicity and ease of confirming tumor implantation; however, metastasis to other organs is rare [11,12]. Transplantation into the subrenal capsule is associated with a high engraftment rate. However, the procedure is difficult to perform, and confirming tumor implantation and growth is challenging. Transplantation into the subrenal capsule of SCID mice is commonly used for prostate PDX models because of the high success rate [69]. Orthotopic transplantation has a higher engraftment rate than s.c. transplantation in one report [70]. Some researchers prefer it because it reproduces the cancer situation exactly. In orthotopic models, the tumor fragments are transplanted in the same anatomical location as they have originally existed. The orthotopic models resemble original human cancers concerning histology, vasculature, gene expression, response to chemotherapy, and metastatic biology. In comparison to s.c. models, orthotopic models generally show the appropriate metastatic pattern associated with each disease. Orthotopic models play an important role in cancer research associated with tumor growth, invasion, metastasis, and microenvironment because they have a similar tumor microenvironment to the primary tumors [71,72]. In CC-PDX, no metastasis occurred in s.c. models. However, orthotopic models demonstrated metastasis, including peritoneal dissemination, liver metastasis, lung metastasis, and para-aortic lymph node metastasis [30].

### 3.4. PDX Procedure and Success Rate

Few reports of CC-PDX models have been published. This may be a clinical feature of cervical cancer. Most patients with advanced-stage tumors undergo biopsy followed by radiation and chemotherapy treatments. Patients with small lesions may be treated surgically at an early stage but obtaining a sufficient specimen for transplantation is difficult. However, the PDX model is advantageous for these patients because tumor tissue can be expanded in mice, and various experimental methods for tumor tissue analysis can be applied. In the current review, the transplantation success rate varies from 0% to 75% [28,29,30,31,32,33,34,35,36,37]. The most important factor may be the characteristics of each primary tumor, such as its invasion and proliferation.

In the published literature about PDX models of other organs, primary tumor burden is associated with the success rate. Tumor fragments obtained from patients with a large tumor size or advanced stage have a higher success rate of transplantation than tumors obtained from patients with less advanced disease [11,47,73,74,75,76]. Tissue fragments obtained from sites of metastasis have a higher success rate than those obtained from the primary site [77,78,79].

### 3.5. Translational Applications of PDX Models

Few reports of CC-PDX models have been published. However, PDX models are promising tools for translational research. PDX models have been used for various types of cancer research such as drug evaluation, precision oncology, biomarker detection for drug sensitivity or resistance, and comprehension of tumor behavior [16,17,80,81]. For example, Berotti et al. found that amplification of the Erb-B2 receptor tyrosine kinase 2 (*ERBB2*) gene is a determinant of driver of cetuximab resistance and predicts the response to epidermal growth factor receptor (EGFR) and HER-2-targeted therapies in a colon cancer PDX model [3]. The results described above were derived from clinical trials [4,82,83]. Other biomarkers for cetuximab resistance were also identified in colon cancer PDX [84,85,86,87,88]. PDX has the potential to replicate cancer heterogeneity among patients better than traditional cell lines for drug discovery and biomarker identification, allowing for real population-scale studies. The large PDX trial format is well suited for accurately predicting clinical trial responses and capturing potential treatments [17]. An “avatar” is defined as a PDX that received the same anticancer agent that the donor patient received [89]. In colon cancer, an avatar trial is ongoing. The most important PDX model recently identified may be the humanized mouse for development of immunotherapy [13]. NK cell-deficient mice, including NSG and NOG, are commonly used as humanized mice. Three types of human immune cells, including peripheral blood lymphocytes (PBLs) [90,91,92], CD34+ cells [93], and bone marrow–liver–thymus (BLT) [94,95,96] are also used. After irradiation of immunodeficient mice, PBLs or CD34+ cells are transplanted intravenously, intraperitoneally, or via another route. A piece of BLT is also implanted into the subrenal capsule of immunodeficient mice that received prior irradiation. Tumor fragments are implanted into humanized mice with a human immune system. In these PDX, the anticancer immune response is investigated.

## 4. Materials and Methods

### 4.1. Protocol and Registration

A systematic review of articles on CC-PDX registered in PubMed (National Library of Medicine) was conducted according to the guidelines of the PRISMA statement [48]. We also searched for a previous or ongoing review on PROSPERO. We searched on PROSPERO using the MeSH terms “Uterine Cervical Neoplasm” and “Xenograft Model Antitumor assay”. However, no previous or ongoing reviews were found. The current review is not registered.

### 4.2. Information Sources and Search Strategies

The search was performed in PubMed. The following search strategy was used for the database: (uterine cervical neoplasm [MeSH Terms]) AND (antitumor assay, xenograft [MeSH Terms]) OR (xenograft [TIAB])). In addition, we manually searched the bibliographies and references of all selected studies to identify publications that were not retrieved by the primary database search. We used Rayyan (Rayyan Systems Inc., Cambridge, MA, USA. http://rayyan.qcri.org, accessed on 7 July 2021) for screening of articles.

### 4.3. Eligibility Criteria

For animal models of CC-PDX, we searched for papers that used human tissue fragments to create mouse xenografts. Xenografts of any number of passages were considered for inclusion. The year of publication was not restricted.

Exclusion criteria included (1) xenografts grafted from established cell lines, (2) previously manipulated in vitro, and (3) conference proceedings, abstracts, commentaries, and reviews.

### 4.4. Study Selection

Initially, two reviewers (T.T. and S.M.) independently screened the titles and abstracts. The process after screening was performed by the reviewers together. The papers that each reviewer selected were gathered and carefully read by the reviewers as full papers to determine if the full text met the inclusion or exclusion criteria. In case of disagreement, we consulted with a third reviewer (M.O.). The reasons for the exclusion of papers in this second-stage screening process were documented.

### 4.5. Data Extraction and Synthesis

The following items were checked in each selected study and recorded on a provided form: (1) author’s name, (2) year of publication, (3) country where the experiment was performed, (4) type of recipient animal, (5) histology of the primary tumor, (6) type of method to obtain the tumor, (7) type of method for transplantation of the tumor fragment, (8) time from surgery to implant, (9) implanted tumor size, (10) engraftment site, (11) method of grafting, (12) time between implantation and tumor establishment, (13) number of donor patients, (14) success rate of PDX establishment, and (15) main objectives of PDX research. Furthermore, if a drug was tested, its target was confirmed. Studies specifically aimed at assessing the methodological aspects of PDX were annotated with validation methods, such as preservation of histology, driver gene mutations, gene expression, copy number polymorphisms, immunohistochemistry, and proteomics.

### 4.6. Quality Assessment

A critical evaluation of the selected articles was performed using the tool created by Collins et al. [36]. The following items were evaluated: (1) statement of ethical approval, (2) apparent description of the model in detail, (3) apparent instructions on routine maintenance of the model, (4) further preparation of the model for an experiment, (5) information about the tracked/proven tissue of origin, (6) confirmation that the xenograft was obtained from a donor patient, (7) histological confirmation that the xenograft is comparable to the primary tumor, and (8) information about concordance between the PDX model and the patient with respect to response to standard therapy. For each item, the included articles were classified as follows: “yes” = bias with low risk, “no” = bias with high risk, “unclear” = bias with unclear risk, or “not applicable.”

## 5. Limitation

The present study has several limitations that cannot be overlooked. First, the sample size was small. Second, the methods and calculation of results were not standardized. For example, the number of mice used, and calculation of the engraftment rate varied. Third, most studies did not include a control group because of the characteristics of PDX research. For these reasons, our results must be confirmed in further investigations.

## 6. Conclusions

In the CC-PDX model, the overall success rate was 62.2%. Severe immunodeficient mice and subrenal capsule implantation led to a higher engraftment rate. However, orthotopic and subcutaneous implantation were alternatives. When using nude mice, subrenal implantation may be better. A tumor fragment size of 1–3 mm^3^ was suitable for CC-PDX. Comparing between the primary tumor and PDX, several aspects, including genomic and histological characteristics and sensitivity to anticancer drugs, were conserved.

## Figures and Tables

**Figure 1 ijms-22-09369-f001:**
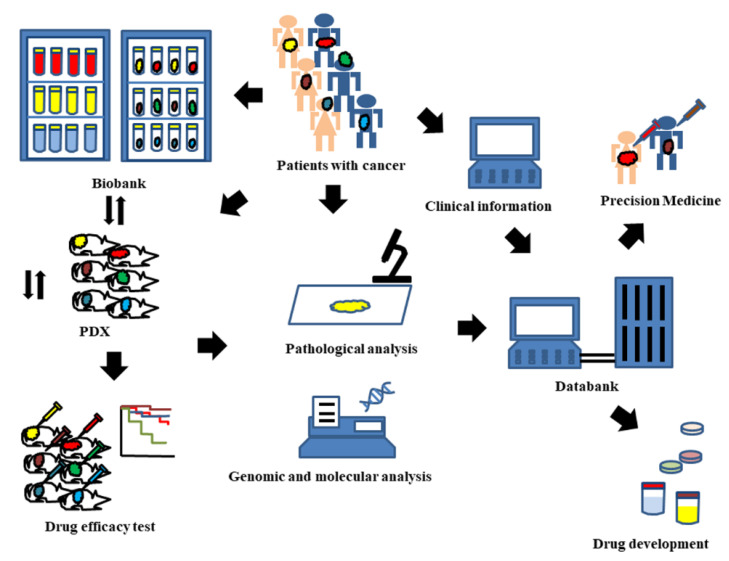
Brief experimental chart for use of patient-derived xenograft (PDX) models. PDX models can be obtained by grafting the tissue resected by surgery or biopsy into immunodeficient mice. All materials and information from cancer patients and PDX models are stored in biobanks and databanks. The drug responsiveness data on cancer patients or PDX models are collected in databanks and intended for use in precision medicine and development of anticancer agents.

**Figure 2 ijms-22-09369-f002:**
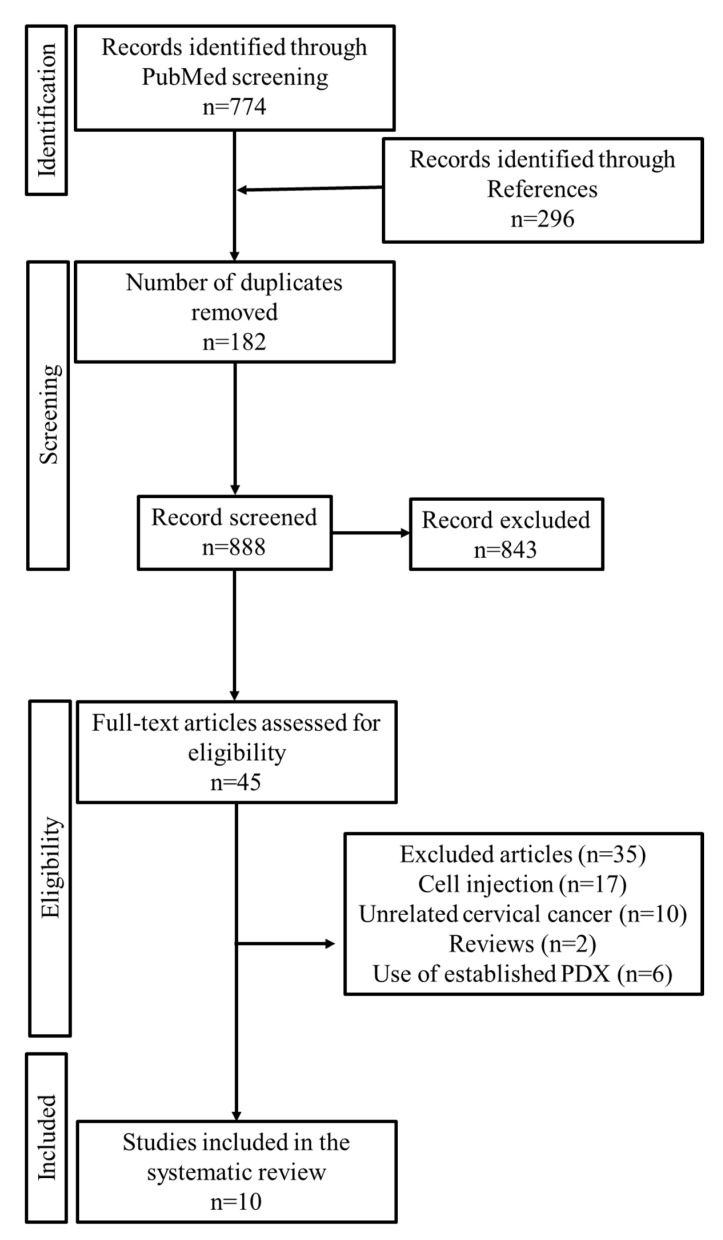
The process of search and selection.

**Figure 3 ijms-22-09369-f003:**
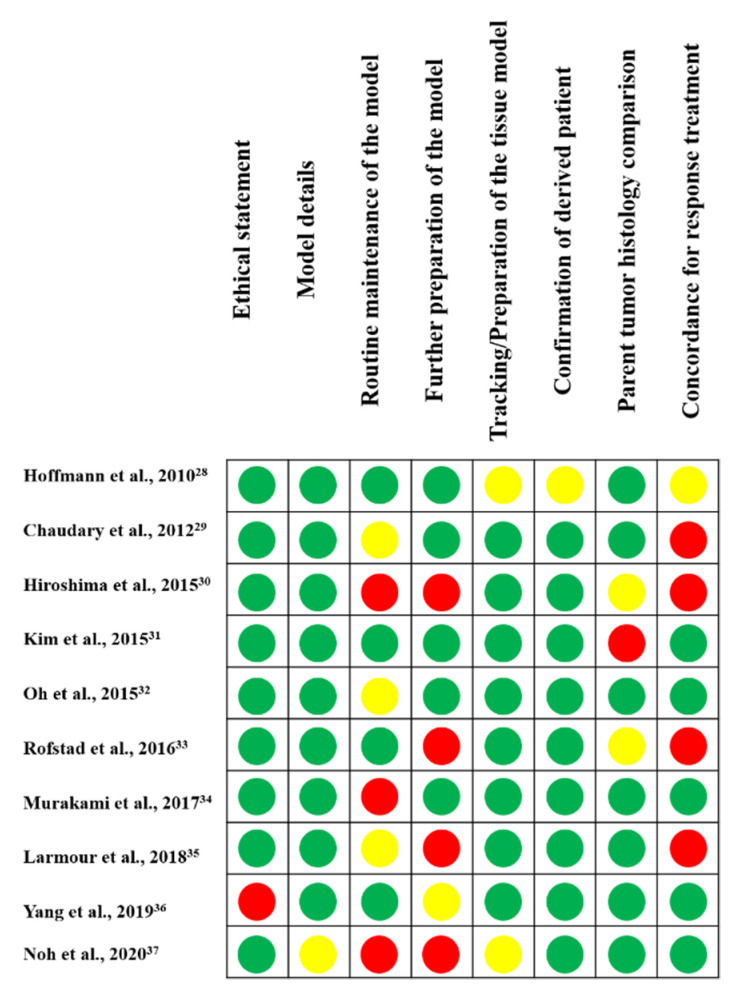
Quality assessment of the studies included in this systematic review. Green circles indicate studies that evaluated the item (low risk of bias), red circles indicate studies that did not show evaluation of the item (high risk of bias), and yellow circles indicate studies that did not present or only partially presented evaluation of the item.

**Table 1 ijms-22-09369-t001:** Information about the selected studies.

Author, Year	Country	Animal Model	Histology	Type of Procedure for Obtaining the Tumor	Time between Surgery and Implantation
Hoffmann et al., 2010 [28]	Germany	SCID	Sq, Ad, Ne	NI	2–5 h
Chaudary et al., 2012 [29]	Canada	SCID, NOD SCID	Sq, Ad, C, M	Biopsy	Immediately
Hiroshima et al., 2015 [30]	Japan	Nude	NI	Surgery	NI
Kim et al., 2015 [31]	Korea	Nude	NI	Surgery	NI
Oh et al., 2015 [32]	Korea	Nude	Sq, Ad	Surgery	NI
Rofstad et al., 2016 [33]	Norway	Nude	Sq	Biopsy	NI
Murakami et al., 2017 [34]	Japan	Nude	Sq	Surgery	Immediately
Larmour et al., 2018 [35]	Australia	NSG	Sq, Ad	Biopsy	<8 h
Yang et al., 2019 [36]	China	NCG	Sq	Biopsy	<12 h
Noh et al., 2020 [37]	Korea	Nude	Sq, Ad	Surgery	NI

SCID, severe combined immunodeficiency; NOD, nonobese diabetic; NSG, NOD SCID gamma; NCG, NOD-Prkdcem26ll2rgem26Nju; NI, no information; Sq, squamous cell carcinoma; Ad, adenocarcinoma; Ne, neuroendocrine carcinoma; C, clear cell carcinoma; M, mucinous carcinoma.

**Table 2 ijms-22-09369-t002:** Information about the selected studies.

Author, Year	Fragment Size	Site of Transplantation	Method of Graft	Mean Latency	Number of Donor Patient	Engraftment Rate (%)	Aim of the Study
Hoffmann et al., 2010 [28]	3–5 mm	Subcutaneous	Direct		6	0 (0/6)	Evaluate the PDX method
	Minced		Injection	6–8 weeks	10	70 (7/10)	
Chaudary et al., 2012 [29]	1–2 mm	Cervix	Direct	3–4 months	33	48 (16/33)	Evaluate the PDX method
Hiroshima et al., 2015 [30]	3 mm^3^	Subcutaneous	Direct	10–15 days	1	70 (7/10)	Evaluate the PDX method
	3 mm^3^	Cervix	Direct	10–15 days		75 (6/8)	
Kim et al., 2015 [31]	2–3 mm	Subrenal capsule	Direct	NI	1	NI	Drug evaluation
Oh et al., 2015 [32]	1 mm^3^	Subrenal capsule	Direct	2–12 months	21	66.7 (14/21)	Drug evaluation
Rofstad et al., 2016 [33]	1 mm	Subcutaneous	Direct	NI	4	NI	Evaluate the PDX method
Murakami et al., 2017 [31]	5 mm	Subcutaneous	Direct	4 weeks	1	NI	Drug evaluation
Larmour et al., 2018 [35]	1 mm^3^	Subrenal capsule	Direct	227 days	14	71.4 (10/14)	Evaluate the PDX method
Yang et al., 2019 [36]	8 mm^3^	Subcutaneous	Injection	73 (21–130) days	2	NI	Drug evaluation
Noh et al., 2020 [37]	2–3 mm	Subrenal capsule	Direct	NI	5	NI	Drug evaluation

PDX, patient-derived xenograft; NI, no information.

**Table 3 ijms-22-09369-t003:** Validation methods and parameters used to demonstrate that PDXs resemble their donor patient tumors in the seven studies that explored the PDX.

Author, Year of Publication	Driver Gene Mutation	Gene Expression	Copy Number Variation	Proteomics	Immunohistochemistry
Hoffmann et al., 2010 [28]	No	No	No	No	EGFR, MHC, p16, ki-67, HPV18E7
Chaudary et al., 2012 [29]	No	No	No	No	CA9, CD31, Ki67, LYVE, EF5, SMA, collagen IV, cytokeratin
Hiroshima et al., 2015 [30]	No	No	No	No	HER-2
Oh et al., 2015 [32]	No	Yes	Yes	No	HER-2
Murakami et al., 2017 [34]	No	No	No	No	No
Larmour et al., 2018 [35].	No	No	No	No	P16, HuNu, HPV, CD45
Yang et al., 2019 [36]	No	No	No	No	No

## Data Availability

The data presented in this study are available on request from the corresponding author.

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
