# Peer review of "Patient-Derived Xenograft Models in Cervical Cancer: A Systematic Review"

_ijms, 2021, doi:10.3390/ijms22179369_

Round 1

Reviewer 1 Report

The signifcance of each PDX method is not well documented in the context of the PDX mouse model that is used for refractory cervical cancer patients.

Author Response

We appreciate the time and effort of the referee in reviewing our manuscript. We have addressed all of the issues indicated in the review report and hope that the revised version meets the journal’s requirements for publication.

Response to Comments from Reviewer 1:

Comment 1:

The significance of each PDX method is not well documented in the context of the PDX mouse model that is used for refractory cervical cancer patients.

Response:

According to your suggestion, we added the sentences about refractory cervical cancer patients. (page 2, line 46-54)

Reviewer 2 Report

The authors made a great effort to revise the manuscript. Actually, the manuscript has been revised well. I have some suggestions to improve the manuscript.

In the revised manuscript, there are several run-on paragraphs that are excessively lengthy. Long winding sentences tend to confuse readers and may lead to misinterpretations. Short sentences are preferred for improved clarity and readability. Therefore, I suggest revising this segment using short sentences.

Abstract
What is the PDX with established cell lines? 
The authors may delete “the overall success rate was 49.5%” in the first sentence of the conclusion.

Introduction
In the introduction, the authors should show the rationale of this study. Moreover, the authors should mention the clinical problems to treat women with advanced cervical cancer.

Results
Why did the authors miss the study reported by Noh J et al (PMID: 33086573)?

Table 1
The author may add the abbreviations of Sq, Ad, Ne, etc.

Figure 2
In the excluded studies, the authors may delete (n=0) criteria.

Figure 3
Please update the reference numbers.

Author Response

We appreciate the time and effort of the referee in reviewing our manuscript. We have addressed all of the issues indicated in the review report and hope that the revised version meets the journal’s requirements for publication.

Response to Comments from Reviewer 2:

Comment 1:

In the revised manuscript, there are several run-on paragraphs that are excessively lengthy. Long winding sentences tend to confuse readers and may lead to misinterpretations. Short sentences are preferred for improved clarity and readability. Therefore, I suggest revising this segment using short sentences.

Response:

According to your suggestions, we revised the long sentences using short ones. (page 1, line 71; page 8, line 208, line231-236; page 9, line 268-270; page 10, line 314)

Comment 2:

What is the PDX with established cell lines? 

Response:

We mean “xenograft using cell lines”.

As you suggested, we revised the sentences. (page 1, line 25)

Comment 3:
The authors may delete “the overall success rate was 49.5%” in the first sentence of the conclusion.

Response:

According to your suggestion, we deleted the sentences. (page 1, line 37)

Comment 4:

Introduction
In the introduction, the authors should show the rationale of this study. Moreover, the authors should mention the clinical problems to treat women with advanced cervical cancer.

Response:

According to your suggestion, we added the sentences about refractory cervical cancer and PDX. (page 2, line 46-54)

Comment 5:

Results
Why did the authors miss the study reported by Noh J et al (PMID: 33086573)?

Response:

We apologize that we missed the study which Noh et al reported.

We added the study in our review.

Comment 5:

Table 1
The author may add the abbreviations of Sq, Ad, Ne, etc.

Response:

According to your suggestion, we added the abbreviations of Sq, AD, Ne, C and M in table 1. (page 5, line 134)

Comment 6:

Figure 2
In the excluded studies, the authors may delete (n=0) criteria.

Response:

According to your suggestion, we deleted the criteria of “n=0” in figure 2. (page 4)

Comment 7:

Figure 3
Please update the reference numbers.

Response:

We apologize the incorrect reference number. We updated them in figure 3. (page 7)

Reviewer 3 Report

Please see in the attached file, thanks!

Author Response

We appreciate the time and effort of the referee in reviewing our manuscript.

Round 2

Reviewer 1 Report

The autor should enphasize the importance and significance of orthotopic xenogarft.  It is quite different from subcutaneous xenograft and others.

Author Response

We appreciate the time and effort of the referee in reviewing our manuscript. We have addressed all of the issues indicated in the review report and hope that the revised version meets the journal’s requirements for publication.

Response to Comments from Reviewer 1:

Comment 1:

The author should emphasize the importance and significance of orthotopic xenograft.  It is quite different from subcutaneous xenograft and others.

Response:

According to your suggestion, we added the sentences and references about orthotopic xenograft model in discussion section. (page 10, line 300-308)

This manuscript is a resubmission of an earlier submission. The following is a list of the peer review reports and author responses from that submission.

Round 1

Reviewer 1 Report

Orthotopic xenograft is very important now.  You should point out it clearly.

Reviewer 2 Report

The authors performed a systematic review about the previous studies that used patient-derived xenograft models in cervical cancer. The authors concluded that cervical cancer PDX models are a promising tool for translational research. The reviewer thinks this study is of some interest; however, I have several concerns regarding this study, especially in the methodology. The reviewer’s comments are as follows.

Abstract

The authors did not show the exact data about the translational research using the PDX models of cervical cancer. Therefore, it is not recommended to conclude that the PDX models of cervical cancer are a promising tool.

The reviewer feels that this abstract is superficial and no useful information. The authors may re-write the entire Abstract.

Lines 49-51: Please add more information about these sentences. Please also cite the previous study.

Line 59: please clarify the “several problems.”

Line 70: Recently PRISMA guideline has been updated; thus, the PRISMA guideline cited by the authors is outdated.

Table 1: the authors may delete the “histology” and “other” column.

Table 1: Please check the legends of Table 1. Unnecessary abbreviations are listed.

Lines 127-236: The authors may move some sections from Discussion to Results.

The authors did not mention the strength and limitations of this study. This study has several notable limitations, and the authors should clarify these points.

Line 240: As mentioned above, the PRISMA guideline is outdated.

The authors report adherence to the PRISMA guidelines. Indeed most of the PRISMA guidelines have been met (i.e. a structured abstract and inclusion of all full search strategies), but the authors have not included the completed checklist to accompany the main text nor do the authors mention the preregistration of the systematic review for instance PROSPERO.

Has an information specialist been involved in the design of the strategy? If so, mention his/her contribution to the methods and acknowledgments.

The authors should use MeSH keywords. Please also use [TIAB] codes.

Please describe any online tools used in the screening process.

Please clarify the inclusion criteria of this study.

The search methods should clarify the general design of the search and any limitations used. Instead of mentioning all synonyms rather explain that the concepts “Uterine cervical cancer” and “PDX model” were searched using text words and MESH-terms in PubMed. Also, mention any limitations used. The authors may list the search terms in detail in a Supplemental Table.

Please mention whether these authors (T.T. and S.M.) independently extracted the data.

Reviewer 3 Report

Please find in the attached file, thanks!
